# Cyclic Photoisomerization of Azobenzene in Atomistic Simulations: Modeling the Effect of Light on Columnar Aggregates of Azo Stars

**DOI:** 10.3390/molecules26247674

**Published:** 2021-12-18

**Authors:** Markus Koch, Marina Saphiannikova, Olga Guskova

**Affiliations:** 1Institute Theory of Polymers, Leibniz Institute of Polymer Research Dresden, Hohe Str. 6, 01069 Dresden, Germany; grenzer@ipfdd.de; 2Dresden Center for Computational Materials Science (DCMS), Technische Universität Dresden, 01062 Dresden, Germany

**Keywords:** azobenzene, photoisomerization, photostationary state, multiphotochromic systems, supramolecular assembly, molecular dynamics, computer simulations

## Abstract

This computational study investigates the influence of light on supramolecular aggregates of three-arm azobenzene stars. Every star contains three azobenzene (azo) moieties, each able to undergo reversible photoisomerization. In solution, the azo stars build column-shaped supramolecular aggregates. Previous experimental works report severe morphological changes of these aggregates under UV–Vis light. However, the underlying molecular mechanisms are still debated. Here we aim to elucidate how light affects the structure and stability of the columnar stacks on the molecular scale. The system is investigated using fully atomistic molecular dynamics (MD) simulations. To implement the effects of light, we first developed a stochastic model of the cyclic photoisomerization of azobenzene. This model reproduces the collective photoisomerization kinetics of the azo stars in good agreement with theory and previous experiments. We then apply light of various intensities and wavelengths on an equilibrated columnar stack of azo stars in water. The simulations indicate that the aggregate does not break into separate fragments upon light irradiation. Instead, the stack develops defects in the form of molecular shifts and reorientations and, as a result, it eventually loses its columnar shape. The mechanism and driving forces behind this order–disorder structural transition are clarified based on the simulations. In the end, we provide a new interpretation of the experimentally observed morphological changes.

## 1. Introduction

One of the most common pathways to create light-responsive material is the incorporation of molecular photoswitches. Such compounds switch between at least two stable (or metastable) states upon external irradiation, significantly changing their properties in the process. In this respect, the chromophore azobenzene (azo) is applied with great success [1]. Due to its remarkable versatility, azobenzene is employed in countless industrial and scientific endeavors. It is embedded into supramolecular assemblies [2,3,4] and proteins [5] to modify the structure of materials and influence biological processes. Polymers incorporating azo moieties can be patterned by irradiation [6,7,8,9,10,11] and azo-based LC-polymers generate strong opto-mechanical responses that can be harnessed for light-controllable actuators [12,13,14], to name only a few examples.

Being perhaps the most widely used molecular photoswitch, the prevalence of azobenzene is substantiated by its outstanding properties. Azobenzene strongly absorbs light, upon which it reversibly isomerizes in “one of the cleanest photoreactions known” (H. Rau) [1,15], i.e., virtually free of side reactions. Thus, azobenzene switches between the stable, planar *trans* and the metastable, non-planar *cis* isomer, accompanied by substantial motion [1,16].

In molecules and materials that combine multiple azobenzene groups, the interactions of the chromophores among each other and with their surroundings play an essential role. For instance, multiphotochromic molecules contain several azo moieties, which are chemically closely connected. Such compounds have a complex isomerization behavior and are promising molecular multi-state systems that can be controlled by light [17,18,19]. Key aspects to consider in multiphotochromic molecules are the typically undesirable electronic coupling between the azobenzene groups as well as the interplay of the azo moieties and other constituents of the system in total. In particular, when the azobenzenes are isomerized by light, they impact their environment with substantial motion and energy dissipation.

Light can induce highly disparate effects in azo materials. It strongly depends on the details of the system, what outcome is generated by external irradiation. On the one hand, it can induce collective molecular reorientation [1,20], leading to directional macroscopic deformations [6,7,12,14,21,22,23]. On the other hand, light can disrupt an existing order [24,25] or erase the structure of a patterned surface [8,10,11,26].

An example of the light-induced disruption of order is found in supramolecular assemblies formed by azo-functionalized star-shaped molecules with a benzene-1,3,5-tricarboxamide (BTA) core [24,25]. In a 1:1 mixture of water and DMSO at room temperature, the azobenzene stars yield needle-like structures. These needles are several tens to hundreds of micrometers in length and 3–5 μm in diameter [24]. Upon irradiation with UV light (λ = 330–385 nm) their structure is transformed and the previously observed needles disappear within about 10s. The morphological transition is reversible, such that after 2min of applying blue light (λ = 460–490 nm) or after 30min in the dark, the needle-like structures are rebuilt. It is possible to repeat the disappearance and renewed appearance of the needle-like structures multiple times without degradation of the material. The authors have demonstrated that the light irradiation generates no noticeable heat. Moreover, heat-induced melting of the structures necessitates temperatures of at least T=70 °C. Systems of this kind are promising precursors of light-controllable materials with self-healing properties. Studying them helps to advance the rational design of smart functional materials such as light-responsive gels [27,28].

Despite long-standing scientific efforts regarding azobenzene and azo-based materials, in many cases, the details behind such light-induced changes are still unclear. For instance, the molecular driving forces leading to the reversible “melting under light” of the needle-like structures [24,25] are still under debate [29]. Thus, in this work, we aim to find out how irradiation with UV–Vis light impacts the stability and shape of supramolecular aggregates formed by the three-arm azobenzene stars. Furthermore, it is our goal to determine the mechanism behind these morphological changes.

So far the formation of stack-like aggregates by BTA groups, which are at the centers of the here considered molecules, has been the focus of numerous previous studies [28,30,31,32,33,34,35,36]. On top of that, several works have investigated supramolecular structures formed by light-responsive azobenzene stars with BTA centers [24,25,37,38]. A detailed simulation study has focused on the introduction of defects in supramolecular tubules of another multiazobenzene compound [39]. We have previously investigated the self-assembly and stability of aggregates formed by three-arm azobenzene stars [40,41] and we have analyzed how highly intense light irradiation influences the binding of these supramolecular stacks [29]. In a proof-of-principle manner, it was found that the photoisomerization of all *trans*-azo groups to *cis* does not result in breaking or disassembly of the column-shaped aggregate but strongly disorders it.

In this article, columnar stacks of three-arm azobenzene stars are investigated using a new and more accurate approach regarding the influence of light. We have developed the so-called cyclic photoisomerization model to implement the collective photokinetics of azo-containing systems in fully atomistic molecular dynamics (MD) simulations. This approach is based on existing simulation models [42,43,44,45,46] but is further extended and refined in this work. As a result, our simulations reproduce the collective photoisomerization kinetics of azobenzene in good agreement with theory and previous experiments—however, on a shorter time scale. The new simulation approach, described in detail in this work, can be highly useful for future computational studies of azo-containing systems. We wish to point out that this article is based on the dissertation of M.K. [47].

The remainder of this work is structured as follows. In the next section, the employed models and methods are presented. There, the three-arm azobenzene star and its columnar aggregates are described in depth. Furthermore, we explain the simulation details, including the cyclic photoisomerization model. Following this, the results of the simulation study are presented and discussed. First, the simulated photokinetics is analyzed and compared to theory and previous experiments. Next, the effects of light on a supramolecular aggregate formed by three-arm azobenzene stars are evaluated regarding structural and energetic changes. It follows an explanation of the mechanism behind the observed order–disorder transition. Finally, conclusions with respect to the research question and the experimental systems are drawn. The Appendix A contains additional figures, a mathematical model of the simulated photokinetics, and definitions of some observables.

## 2. Materials and Methods

### 2.1. Object of Study

The study focuses on supramolecular aggregates formed by a $C_{3}$-symmetrical multiphotochromic molecule, which we refer to as *TrisAzo* (Figure 1). This star-shaped compound consists of three azobenzene groups, which are centrally connected by a flat BTA group. In addition, each azobenzene arm is para-substituted with a dimethylamino (DMA) group, R = N(CH_3_)_2_. Whereas the BTA and dimethylamino groups are photoinert, each azobenzene arm can undergo reversible *trans*–*cis* photoisomerization upon absorption of a photon with a suitable wavelength. In addition, the *cis* isomers of azobenzene may switch back to *trans* due to thermal relaxation. Thus, *TrisAzo* has four different isomerization states, ranging from the *trans*–*trans*–*trans* isomer (*ttt* or all-*trans*), over two mixed isomers, to the *cis*–*cis*–*cis* isomer (*ccc* or all-*cis*), see Figure 1. We have investigated various molecular properties (solvation, light absorption, etc.) of *TrisAzo* depending on its isomerization state in a previous computational study [48].

Note that the three-arm azo star investigated in Ref. [24] has hydrogen atoms (R = H) in place of dimethylamino groups and will be named *TrisAzo-H* from here on. These substituent groups have only a marginal influence on the intermolecular interactions between such molecules [41].

In another work, we have investigated the self-assembly behavior of *TrisAzo* molecules in water [29]. We verified that *TrisAzo* molecules form stack-like aggregates. In this work, we therefore consider a pre-assembled stack of *TrisAzo* with an initial stacking distance of 4Å and N=36 molecules (Figure 2a). In the particular cluster arrangement considered here, every second monomer in the stack is flipped by 180°. Each monomer is placed on top of the other with aligned centers and azo groups. This arrangement leads to the best possible stacking of the BTAs and the azo groups. Moreover, it enables the formation of a triple H-bonding pattern between neighboring BTAs due to the favorable placement of the amide groups of the BTAs.

This initial stack arrangement is considered because it has been the most long-time stable in the columnar shape, as determined in a closely related simulation study of the equilibrium structure of *TrisAzo* stacks [41]. Thus, after the equilibration period, the simulations can be continued with a still column-shaped stack (Figure 2b). More details about the structure of the equilibrated stack are given in Section 3.2.1.

### 2.2. Basic Simulation Model

To study the system, fully atomistic MD simulations are employed. All simulation runs are carried out in the open-source simulation software LAMMPS [52,53]. The *TrisAzo* molecules are parameterized in the DREIDING force field [54]. Note that this force field contains a hydrogen-bonding potential term. Thus, we can explicitly model the formation of hydrogen bonds between the amide groups in the BTAs at the center of the *TrisAzo* molecules. In particular, nitrogen acts as the hydrogen donor and oxygen as the acceptor (N–H⋯O). We use the standard parameters of the H-bonding potential of DREIDING [54]. Assignment of the atom types and Gasteiger partial charges [55] is aided by the molecular modeling program BIOVIA Materials Studio [56]. Pre-assembled *TrisAzo* clusters are prepared and solvated in SPC/E water [57] using the free software packages Moltemplate [58] and PACKMOL [59,60]. The resulting configurations are then transferred to LAMMPS.

The cutoff radius for the LJ potential is set to 9.8 *Å* and LJ interactions between unlike atom types are obtained via Lorentz–Berthelot combination rules [61,62]. Coulomb interactions are evaluated via the PPPM algorithm with an accuracy of $10^{-4}$ [63,64]. Thermostatting, barostatting, and time-integration are carried out via a Nosé–Hoover-style algorithm as implemented in LAMMPS [52,53]. Temperature and pressure are kept at constant average values of T=300K and P=1atm (NPT ensemble), respectively, to match the experimental conditions of Ref. [24]. The lengths of covalent bonds that involve hydrogen atoms and their associated bond angles are constrained via the SHAKE algorithm [65]. The time step of the simulations is 1 fs. The pre-assembled and solvated clusters are equilibrated for 35 ns before the effects of light are applied (see below). Equilibration is established based on stable pair and total energies in the system as well as unchanging structural properties of the cluster.

Visualizations of the molecular structures generated in LAMMPS are created in the open-source 3D computer graphics software Blender [49]. The MD data is transferred to this program using the Blender plugin BlendMol [50] and the molecular visualization program VMD [51].

### 2.3. Modeling the Photoisomerization of Azobenzene

To simulate individual photoisomerization events of azobenzene in fully atomistic MD simulations, the equilibrium force field needs to be modified, i.e., a reactive mechanism needs to be introduced. Thus, switching between the *trans* and *cis* geometry of azobenzene can be incorporated. Out of multiple reported MD implementations of the photoisomerization reaction [39,42,43,66,67,68,69,70,71,72,73], we employ the model of Heinz et al. [42]. It has been used in various other simulation works [39,42,73,74,75,76,77] and is consistent with the most important features of the isomerization of azobenzene. Furthermore, the model is directly compatible with a range of force fields, among them the DREIDING force field employed in this study.

The approach incorporates the so-called rotation pathway of photoisomerization [16]. It is realized by temporally changing the dihedral potential of the atoms C-N=N-C at the center of azobenzene. The model provides three sets of parameters for this dihedral potential [42]: One for the equilibrium state of azobenzene, and two for modeling the transitions from *trans*→*cis* or *cis*→*trans*, respectively. During the switching from one isomer to another, the respective dihedral potential is temporally activated. In our study, this activation lasts 1.5 ps, consistent with the original model [42]. This is sufficient to generate the new geometry. After this step, the equilibrium dihedral potential is activated again. In the selected implementation, which is compatible with the DREIDING force field, the dissipated energy is 78 $\mathrm{kcal}\cdot {\mathrm{mol}}^{-1}$ in both switching directions. The model does not take into account the energy difference of the two isomers. More details of the model (functional forms, force field parameters, etc.) are provided in Ref. [42]. Note that differences of the partial charges and LJ parameters of *trans* and *cis* azobenzene are neglected.

### 2.4. Modeling the Collective Photoisomerization Kinetics: The Cyclic Photoisomerization Model

To perform computer simulations of a system containing multiple azobenzene moieties under irradiation with UV–Vis light, it is not sufficient to implement the photoisomerization mechanism of azobenzene itself. On top of that, the approach needs to model the photoisomerization kinetics of multiple azobenzene groups. The simulations should be able to reproduce cyclic *trans*–*cis*–*trans* photoisomerization as well as reversible photoisomerization in general. This is required because UV–Vis light typically induces both directions of photoisomerization [1,78,79] but in different proportions. The ratio between the two reaction rates depends on the wavelength of the light [79]. Moreover, the system should reach the so-called photostationary state (PSS) upon continued cyclic photoisomerization. The PSS is a dynamic equilibrium in which the conversion rates between the two isomers are equal and the coexisting molar fractions of *trans* and *cis* azobenzene remain at stable values.

We neglect possible heating effects due to the external irradiation with light. In the experimental works which have inspired this investigation [24,25], the samples are not significantly heating up during the application of light. The observed structural changes must therefore directly result from the photoisomerization of azobenzene.

Dynamic simulation models implementing cyclic *trans*–*cis*–*trans* photoisomerization of azobenzene have been developed for coarse-grained [79,80] and fully atomistic simulation studies [43,44,45,46,81] as well as combinations of such approaches [82]. As a starting point, the present investigation adapts the so-called cyclic photoisomerization model (CPM) in close agreement with the protocol of Ref. [43]. On top of that, an extended version of this approach is devised. This modified variant is intended to capture the photoisomerization kinetics of azo-containing systems more realistically and accurately.

#### 2.4.1. Adaptation of the Model by Bedrov et al. (Bedrov Model or CPM-B)

This simulation protocol is adapted from Bedrov et al. [43] and will be further named Bedrov model or CPM-B. Ref. [43] cites the biased simulations of azobenzene materials by Teboul et al. [44,46], on which the time scales and other details are based. The simulations start from an initial configuration equilibrated at experimental conditions (T=300K, P=1atm) in the NPT ensemble. The configurations consist of two components: the azobenzene groups and their environment, i.e., all the photoinert components of the system.

Initially, any light irradiation is absent. Thus, all azobenzene groups are present as *trans* isomers (Figure 3, left-hand side). The simulation then proceeds with a succession of four simulation steps (steps 1–4 in Figure 3), which are cyclically repeated (or looped) for *N* iterations. The steps are as follows:(a)At the beginning of every cycle, a predefined fraction of the *trans*-azo groups is randomly selected for photoswitching. A *trans*-azo moiety is selected with the probability $p\left(\mathrm{trans}\to \mathrm{cis}\right)\equiv p_{tc}=\mathrm{const}.$ This fraction represents those *trans* isomers, which absorb a photon and subsequently undergo photoisomerization. Note that in the original model, irradiation with linearly polarized light is simulated. To implement this, the switching probability $p_{tc}$ should depend on the angle θ between the long axis of the azo moiety and the polarization direction of the incoming light. In particular, this probability should be proportional to $\cos^{2}\theta$ [79]. The probability for the *cis*→*trans* back reaction is assumed to remain angle-independent [79]. After implementing such an angle-dependence, the photo-orientation effect of azobenzene can be realized [43,79]. Here, the switching probabilities remain angle-independent since we do not simulate light of a certain polarization.(b)For the subset of azo groups now selected for switching, the photoisomerization model of Heinz et al. [42] is applied (see Section 2.3). To this end, the C-N=N-C dihedral potential of the selected azobenzene groups is temporally modified. The simulation is then continued for $\Delta t_{tc}=1.5\;ps$, allowing the isomerization reaction to complete.Before the simulation enters the next stage, the dihedral potential parameters of the isomerized azobenzene groups are reverted to their equilibrium values. The *cis* isomers remain stable because thermal relaxation back to the *trans* geometry is energetically hindered. The simulation then proceeds for $\Delta t_{cis}=15\;ps$. During this period the system can relax, now containing a mixture of *trans* and *cis* isomers.(a)Next, a fixed fraction of the *cis* isomers is selected for isomerization back to *trans*. *Cis* groups are selected with the probability $p\left(\mathrm{cis}\to \mathrm{trans}\right)\equiv p_{ct}=\mathrm{const}.$ In this implementation (CPM-B), this probability is $p_{ct}$ = 100%. In other words, all *cis* isomers will be switched back to *trans*.(b)The C-N=N-C dihedral potential of the *cis*-azobenzene groups is temporally modified again, now using the parameters for the back reaction [42]. The simulation continues for $\Delta t_{ct}=1.5\;ps$ to complete the conversion.Finally, the modified dihedral potentials are reset to their equilibrium settings. Now all azobenzene groups are again present as *trans* isomers, like in the initial stage. They remain like this for $\Delta t_{trans}=30\;ps$, allowing the system to relax before another iteration of the loop begins.

Despite the detailed setup of this approach, it is characterized by several simplifications, some of them oversimplifying the simulated phenomenon. First, the back conversion of all *cis*-azo groups during each cycle is an artificial element of the model and highly overestimates the probability of *cis*→*trans* isomerization reactions. Second, the system can never reach the PSS since the fractions of *trans* and *cis* azobenzene do not change over time. Lastly, because only short periods of simulated time (tens of nanoseconds) are feasible in fully atomistic MD, the isomerization rate of azobenzene has to be strongly increased to reproduce effects that take seconds or minutes to appear in experiments [24]. Nonetheless, simulations using this model can be insightful as proof-of-principle studies and enable studying the effects of light on azo-materials in a qualitative manner.

#### 2.4.2. The Extended Model (CPM-E)

To improve some of the shortcomings of the CPM-B, a modified variant of the protocol presented above is devised. This version is named extended model (CPM-E). The main change in the modified approach is that the *cis*→*trans* photoisomerization now proceeds with probabilities $p_{ct}\leq$ 100%. Thus, the *cis*-azo groups are generally not all switched back within the same cycle in which they were generated. As a consequence, it becomes possible for the *cis* isomers of azobenzene to accumulate over time.

Note that the CPM-B is a special case of the CPM-E with $p_{ct}$ = 100%. For $p_{ct}<$ 100%, the number of *trans* and *cis* isomers changes at each iteration, in contrast to the CPM-B. Moreover, with $p_{tc}$ = 100% and $p_{ct}$ = 0% it is possible to realize a “full isomerization” case, i.e., simultaneous *trans*→*cis* switching of all azo groups without reversal, as we have studied in a previous work [29].

The proposed modification amends several aspects of the original model. After a given time, a CPM-E simulation can reach the PSS. In this state, detailed balance is established between backward and forward switching. Furthermore, an effect similar to tuning the wavelength of the light source can be incorporated by changing the ratio $p_{tc}/p_{ct}$ [79]. This can be used to generate different PSS distributions of the *trans* and *cis* isomers. A higher ratio, for instance, corresponds to a lower wavelength, i.e., in the UV range, *trans*→*cis* switching dominates. At fixed values of $p_{tc}/p_{ct}$, increasing the value of $p_{tc}$ has an effect comparable to increasing the light intensity. Despite these improvements, some inaccuracies remain in this variant of the model. To feasibly perform the simulations, both versions of the CPM have to use very high photoisomerization rates as compared to real systems.

It should be mentioned that the CPM-B approach remains interesting for simulational studies if one aims to avoid the accumulation of *cis* isomers over time. While this is a rather artificial scenario, it enables the study of azo-containing systems under special conditions: The photoisomerization reactions then mainly cause energy dissipation and motion, whereas lasting changes in the molar fractions of the *trans* and *cis* isomers are suppressed.

## 3. Results and Discussion

### 3.1. Photoisomerization Kinetics

We apply the two variants of the cyclic photoisomerization model on the equilibrated columnar stack of *TrisAzo* molecules to study the effects of light on this system. Note that these are non-equilibrium simulations, as ongoing photoisomerization events can only occur under a steady influx of energy, provided by the incident light.

#### 3.1.1. Photoisomerization Kinetics in the CPM-B Approach

In the CPM-B approach, *cis* azobenzenes cannot be accumulated in the system. Thus, the fractions of the *trans* and *cis* isomers do not undergo any lasting changes and the PSS is never reached. Furthermore, in this approach, the rate of *trans*→*cis* switching on average remains constant throughout the simulation and is equal to the rate of *cis*→*trans* switching. These properties are contradictory to expectations and experimental results [83]. Thus, we omit a further examination of the photoisomerization kinetics in CPM-B simulations.

#### 3.1.2. Photoisomerization Kinetics in the CPM-E Approach

Figure 4 presents the time evolution of the molar fraction (or number fraction) of the *trans*-azo isomers. This molar fractions is given in relation to all azo groups in the system. The results confirm that in the CPM-E approach, the system eventually reaches a state with time-stable fractions of the *trans* and *cis* isomers, i.e., the PSS. As expected, higher switching probabilities cause faster progress towards this state. The same effect is observed for the *TrisAzo* isomers (Figure 5). Recall that increasing $p_{tc}$ at a fixed ratio $p_{tc}/p_{ct}$ corresponds to increasing the light intensity [79].

Furthermore, the progression towards the PSS and the steady-state isomer fractions depend on the ratio of the switching probabilities, $p_{tc}/p_{ct}$, which is related to the wavelength of the light. For the here considered system, the case $p_{tc}\gg p_{ct}$ corresponds to irradiation with UV light like in Ref. [83] (detailed comparison see below). The case $p_{tc}=p_{ct}$ resembles incoming light of a larger wavelength [79]. For equal forward and backward switching rates, the mixed isomers of *TrisAzo* dominate, while the *ttt* and *ccc* isomers become sparse (Figure 5a). If the forward switching rate $p_{tc}$ is much larger than $p_{ct}$, the *ccc* isomers become the dominant moieties and the *ttt* isomers are fully annihilated (Figure 5b). Before the PSS is reached, the initially present *ttt* isomers are first converted to *ttc* and *tcc*, which explains the time-shifted peaks of their curves. Small fractions of such mixed isomers can persist due to the nonzero backswitching probability.

#### 3.1.3. Comparison of CPM-E Results with Theory

To validate the MD data, we compare the simulated photoisomerization kinetics to theoretical results. Previous works have reported detailed analytical models for the photoisomerization kinetics of azo-containing systems [78,79,84]. Here, we assume the same simplified premises for the collective photoswitching as in the simulations. For instance, the thermal back relaxation of the isomers from *cis* to *trans* is not treated as a separate phenomenon. This is done since this effect typically occurs on much larger time scales than the light-induced switching [1]. Moreover, if the light intensities are sufficiently high, the light-induced photoisomerization reactions dominate and the thermally induced reactions occur much more rarely.

The photoisomerization kinetics of the azo groups can be described by two coupled linear ordinary differential equations (ODE). In addition, the interconversion of the four isomers of *TrisAzo* is captured by four coupled ODE. The mathematical models for both photokinetic processes are described in Appendix A. There, we derive closed solutions for both systems of ODE and discuss the initial value problem with 100% *trans* isomers at time t=0. Note that a corresponding mathematical description of the interconverting *TrisAzo* isomers has been derived in the Appendix A of Ref. [83]. In the cited work, the model was used to extract the reaction rate constants of the interconverting isomers from experimental data.

Next, we consider the system in the PSS. The steady-state molar fractions of the *trans* and *cis* isomers of azobenzene can be obtained straightforwardly from the photokinetic model, as shown in the Appendix A. Alternatively, the same results can be obtained through statistical considerations, as presented here. As in the simulation model, we assume a fixed number of azo groups in total. These can be present in two different states (*trans* and *cis* isomers), with the molar fractions $x_{t}$ and $x_{c}$, respectively. Interconversion of the isomers occurs with the constant switching probabilities $p_{tc}$ and $p_{ct}$ via independent, random isomerization events. Under these conditions, the system in the PSS follows the well-known principle of detailed balance,
(1)$$\begin{array}{c} x_{t}^{\mathrm{PSS}}p_{tc}=x_{c}^{\mathrm{PSS}}p_{ct}, \end{array}$$
where $x_{t}^{\mathrm{PSS}}$ and $x_{c}^{\mathrm{PSS}}$ are the molar fraction in the PSS. Using $x_{t}\left(t\right)+x_{c}\left(t\right)=1$ (valid for all *t*), the *trans* and *cis* isomer fractions in the PSS can be written as
(2)$$\begin{array}{c} x_{t}^{\mathrm{PSS}}=\frac{p_{ct}}{p_{tc}+p_{ct}}=\frac{1}{\frac{p_{tc}}{p_{ct}}+1},\;\;\;\;x_{c}^{\mathrm{PSS}}=\frac{p_{tc}}{p_{tc}+p_{ct}}=\frac{\frac{p_{tc}}{p_{ct}}}{\frac{p_{tc}}{p_{ct}}+1}. \end{array}$$

Thus, the steady-state molar fractions of the two isomer types depend only on the ratio $p_{tc}/p_{ct}$ of the switching probabilities. As expected, for $p_{tc}>p_{ct}$ the *cis* isomers dominate in the PSS and vice versa in the opposite case. For $p_{tc}=p_{ct}$ the steady-state fractions of both isomers are equal.

The calculated steady-state molar fractions are in excellent agreement with the simulation results, see Figure 4. The time dependence of $x_{t}\left(t\right)$ is reasonably well reproduced by the photokinetic model presented in the Appendix A. Deviations of the MD data from the theory predictions can be explained by fluctuations due to the small sample size (108 azo groups in the simulations) and the stochastic process of photoswitching.

Let us now consider the fractions of the *TrisAzo* isomers in the PSS. This information can be again obtained from the steady-state solution of the respective photokinetic model. As an alternative, the same expressions can be derived using statistical arguments, as shown below. Again assuming random and independent switching events, the steady-state molar fractions of the *TrisAzo* isomers—$x_{ttt}^{\mathrm{PSS}}$, $x_{ttc}^{\mathrm{PSS}}$, $x_{tcc}^{\mathrm{PSS}}$, and $x_{ccc}^{\mathrm{PSS}}$—are expected to follow a binomial distribution. The reason is that each of the three azo groups in an individual molecule must be in one of two states. The azo group is either present as a *trans* isomer or as a “non-*trans*” isomer, which can only be *cis*. Thus, each *TrisAzo* isomer contains *k**trans*-azo groups and 3−k
*cis*-azo groups, which we denote as $t^{k}$ and $c^{3-k}$. The steady-state molar fraction $x_{t^{k}c^{3-k}}^{\mathrm{PSS}}$ of a particular *TrisAzo* isomer type can therefore be expressed by $x_{t}^{\mathrm{PSS}}$ and $x_{c}^{\mathrm{PSS}}$:(3)xtkc3−kPSS=3kxtPSSkxcPSS3−k=3kpctkptc3−kptc+pct3≡3!(3−k)!k!pctkptc3−kptc+pct3.

Here, Equation (Equation 2) has been used. We find that the steady-state molar fractions of *TrisAzo* depend only on $p_{tc}$ and $p_{ct}$, like for the individual *trans* and *cis* isomers.

The PSS results are in excellent agreement with the simulations (Figure 5). On top of that, the photokinetic model provided in the Appendix A captures the time dependency of the molar fractions with good agreement to the simulation results. As mentioned above, deviations from the predicted curves are reasonably small and result from the small sample size as well as the stochastic nature of the simulation approach.

#### 3.1.4. Comparison of CPM-E Results with Experiments

The time-dependent photokinetics of closely related isomers has been studied in experiments of Kind et al. [83]. Note that this work has involved *TrisAzo-H* molecules which are almost identical to *TrisAzo* except for the substituent groups at the azo arms (see Section 2.1). *TrisAzo-H* has the equivalent four isomer types (from ttt to ccc, compare Figure 1). The molecules are solvated in DMSO and exposed to constant irradiation with UV light (λ=365 nm, $I=0.65\;mW\cdot cm^{-2}$, T=300 K). The experimental system is, thus, very similar to the simulated one.

The molar fractions of each isomer type were detected using ^1^H-NMR spectroscopy. In the experiments, the PSS is reached only after several hours. This is in strong contrast to the here simulated periods, where 500 photoisomerization cycles correspond to 24 ns. The comparison highlights how much the simulation approach has to overestimate the switching probabilities to reach a similar state of the system.

The experimentally determined molar fractions of the *ttt*, *ttc*, *tcc*, and *ccc* isomers in the PSS are $x_{\mathrm{ttt}}^{\mathrm{PSS}}=0.00$, $x_{\mathrm{ttc}}^{\mathrm{PSS}}=0.01$, $x_{\mathrm{tcc}}^{\mathrm{PSS}}=0.16$, and $x_{\mathrm{ccc}}^{\mathrm{PSS}}=0.83$, respectively [83]. These results are in excellent agreement with CPM-E simulations using switching probabilities with a ratio of $p_{tc}/p_{ct}=15\frac{2}{3}$ (Figure 5b). When comparing Figure 5b and Figure 4 of Ref. [83], even the time-dependent progression of the curves agrees very well qualitatively.

The ratio $15\frac{2}{3}$ of the two switching probabilities has been extracted from the same experimental results [83], as described next. Because the fraction of the *trans* and *cis*-azo groups in each type of *TrisAzo* (and *TrisAzo-H*) isomer is known, the steady-state molar fractions of *trans* and *cis* azobenzene in the experimental system can be calculated. We find $x_{t}^{\mathrm{PSS}}=0.06$ and $x_{c}^{\mathrm{PSS}}=0.94$, agreeing well with Figure 4b. Using Equation (Equation 1), *trans*→*cis* isomerization in this case has a $15\frac{2}{3}$ times higher probability than backward switching. This back switching probability implicitly comprises both the light-induced and the thermally induced *cis*→*trans* switching. Ref. [83] demonstrates that the latter occurs on much longer time scales than the former (several days vs. several hours). It is therefore reasonable to neglect thermal back relaxation as another explicit effect in the simulations.

Note also that using Equation (Equation 3) with $x_{t}^{\mathrm{PSS}}=0.06$ and $x_{c}^{\mathrm{PSS}}=0.94$, one finds virtually the same steady-state molar fractions of the *TrisAzo-H* isomers as in the experiments. This apparently trivial result makes a strong case for the assumption that the photoswitching events of the azo groups occur in good approximation randomly and independently—in particular, independently of the current isomerization state of the molecules. Thus, possible coupling effects in this type of multiphotochromic azobenzene star appear to be small. In addition, Kind et al. report that the thermal back relaxation of the *TrisAzo-H* molecules from the PSS indicates that the three azo arms behave like independent photoswitches [83]. The rate constants of the back conversion reactions *ccc*→*ttc*, *tcc*→*ttc*, and *ttc*→*ttt* have the proportions 3:2:1. This implies that the number of *cis* arms in these isomers (3, 2, or 1) determines the rate constants, without any obvious coupling effects between the connected azo groups.

The simulation results demonstrate that the CPM-E reproduces the photoisomerization kinetics of *TrisAzo*-like molecules with good agreement to the experiments of Kind et al. [83]. The MD data correspond well to the mathematical model of the photokinetics—both regarding the time-dependent behavior and the steady-state solutions. The CPM-E therefore replicates essential features of a system of *TrisAzo* (or *TrisAzo-H*) molecules under UV–Vis irradiation. The simulation model is not taking into account possible changes in the isomerization behavior due to the environment of the molecules, e.g., if the *TrisAzo* molecules are tightly packed in a columnar cluster. In such a case, the ability of an azo group to be isomerized may depend on the available free volume in its vicinity. A simulation model as devised in Ref. [39] could amend this.

### 3.2. Cluster Structure before and during Light Irradiation

After verifying that the simulations reproduce the expected photoisomerization kinetics reasonably well, we investigate the impact of light on the structure of the *TrisAzo* stack. To this end, both CPM variants are utilized. Depending on the induced changes in each model, one may conclude which of them is yielding defects in the cluster more effectively. Are the switching events, i.e., primarily their motion and energy dissipation, causing the majority of defects? Then CPM-B and CPM-E simulations with comparable average switching rates should yield similar structural changes in the cluster. Or is the accumulation of *cis*-azo groups an intensifying—possibly even necessary—factor for the generation of defects? In this case, CPM-E simulation should generate defects and other deviations in the stack more strongly.

Note that cluster structure is analyzed using various observables and a defect detection algorithm, which are described in detail within Appendix A.

#### 3.2.1. Equilibrium Structure of the Cluster (Initial State)

The equilibrium structure of the *TrisAzo* is analyzed in detail in another work [41]. Nevertheless, the most important structural properties are summarized below. Thereby, we characterize the reference state of the system before the application of the cyclic photoisomerization model. The structure of the stack is evaluated during the final 25 ns of the total equilibration time of 35 ns.

The cluster remains in a well-ordered stacked arrangement and retains its columnar shape, as indicated by Figure 2b. The average pairwise center-of-mass (COM) distances between adjacent monomers have a value of 〈Δr〉=3.69±0.36 Å. This stacking distance is close to the initial value of 4 Å and has a low standard deviation, thus, indicating highly regular stacking. Moreover, the monomers retain a high degree of orientational alignment, i.e., only small inclination angles to their neighbors are observed. The average cosine of the inclination angles of adjacent monomers is 〈Ψ〉=0.99 with a 68% confidence interval much narrower than 0.01. This corresponds to a nearly perfect columnar stacking order between the monomer pairs. Only the monomers at the ends of the columnar stack are slightly shifted and misaligned, which is substantiated by the defect detection algorithm. This is caused by the weaker binding energies at this position. The terminal monomers have only one binding partner, as compared to the two-sided binding of the inner monomers. The defect detection algorithm does not find any defects inside of the equilibrated *TrisAzo* stack.

In total, the neighboring molecules have regularly stacked BTA groups at the centers and azo groups at the periphery. This enables the amide groups of the opposing BTAs to form a regular triple hydrogen-bonding pattern between each adjacent pair, i.e., the highest possible number of H-bonds between BTA pairs [30,32,35]. The hydrogen bonding additionally contributes to the stability of the columnar shape of the stack.

#### 3.2.2. Cluster Structure upon Application of the CPM-B Approach

Let us follow the visually detectable structural changes of the *TrisAzo* stack in the CPM-B simulations, see Figure 6a. Here, the well-ordered stacking between the azo groups of adjacent molecules is only broken during the short *cis* periods within an isomerization cycle because the bent *cis*-azo groups cannot remain stacked. Upon switching back to the flat *trans* geometry, most azo arms recover their stacked arrangement, especially during the early simulation stages. The cyclic isomerization only sparsely leads to misaligned azo arms. At these positions in the stack, occasionally singular defects start to develop. In particular, by defects we mean adjacent pairs of monomers, in which the stacking is disturbed due to lateral shifting or molecular reorientation. Eventually, the defects lead to the compartmentalization of the cluster into long straight sections. These compartments do not detach or break away from the rest of the cluster but are often strongly tilted towards each other. Finally, the monomer pairs between which defects have developed become significantly displaced and misaligned with respect to their previous binding partners. Over time, an increasing number of defects develop and the remaining straight sections become shorter.

These structural changes arise with different severity depending on the switching probability per cycle, $p_{tc}$. At the lowest considered value ($p_{tc}$ = 1.00%), the structure of the column essentially remains unaltered throughout the considered period. Even the number of defects is not changed. Only a pre-existing defect at the periphery of the stack is detected (Figure 7). For higher values of $p_{tc}$, the columnar orientation parameter Ψ(t) decreases (Figure 8a). Simultaneously, the average distances between neighboring monomers, Δr(t), increase over time (Figure 8c). These findings verify the visible structural degradation of the stack. New defects along the stack are generated accordingly, see Figure 7. The structural changes set in after approximately 100 to 200 photoswitching cycles for $p_{tc}\geq 2.0$. As expected, the final degradation of the cluster structure (after 500 cycles) increases with $p_{tc}$, i.e., with stronger light intensity.

#### 3.2.3. Cluster Structure upon Application of CPM-E Approach

In the case of the CPM-E simulations, the MD snapshots (Figure 6b) reveal a fast accumulation of *cis*-azo groups, in agreement with Figure 4. The continued existence of *cis* azobenzene persistently disrupts the stacking between the azo arms of neighboring monomers due to the bent shape of the *cis*-azo groups. After only a few isomerization cycles, the central BTA groups of some adjacent *TrisAzo* pairs become misaligned and begin to move apart laterally. Thus, the cluster develops defects at these positions, which increase in number and severity over time. Finally, the overall shape of the cluster is deformed and becomes bent instead of linear. The stack then consists of short sections of aligned monomers, which are discontinued by defects. Again, we find no breaking or disassembly of the cluster into detached fragments. One of the main differences between the two CPM variants is that the CPM-E approach generates structural defects earlier and the cluster appears to become fragmented into shorter sections.

In the CPM-E simulations, two different ratios $p_{tc}/p_{ct}$ are considered, as before: $p_{tc}=p_{ct}$ and $p_{tc}=15\frac{2}{3}p_{ct}$. Under these constraints, the probabilities are varied between $p_{tc}$ = 0.100% and 5.000% as well as $p_{tc}$ = 0.188% and 9.400%, respectively (with according variation of $p_{ct}$). In the case of $p_{tc}=p_{ct}$ = 0.100% and $p_{tc}$ = 0.188%, $p_{tc}$ = 0.012%, i.e., the overall lowest of the selected values, the cluster structure remains nearly unchanged over the considered period. Slight changes in the monomer alignment and pair distances appear only at the very end of these simulations (Figure 8b,c).

For higher switching probabilities, Δr(t) increases and Ψ(t) decreases significantly over time. This is in line with the visually detected structural changes of the clusters. The degradation of the stack structure in the CPM-E simulations generally increases with larger switching probabilities (light intensities). Defects in the cluster become more frequent accordingly, though their number is not higher than in the CPM-B systems for the considered simulation parameters (Figure 7). Interestingly, the cluster structures are changing to a similar degree in both CPM variants despite the much higher number of photoisomerization events in most CPM-B simulations, implying higher average switching rates and energy dissipation (Figure 9). However, in accordance with the MD snapshots, the main difference is that the cluster structure degrades very early during the CPM-E simulations—mostly before or around 100 cycles.

In the CPM-E simulations, also the backswitching probability per cycle, $p_{ct}$, affects the structural changes. As an example, for some the largest considered switching probabilities ($p_{tc}=p_{ct}$ = 5.0% and $p_{tc}$ = 9.4%, $p_{tc}$ = 0.6%), the structural changes depend on the ratio $p_{tc}/p_{ct}$. In this particular case, the degradation of the stack is more severe for $p_{tc}=p_{ct}$ than for $p_{tc}=15\frac{2}{3}p_{ct}$. It should be noted that the average rate of switching events over the course of the simulation, and hence, the amount of dissipated energy, is much higher for $p_{tc}=p_{ct}$ (Figure 9). Furthermore, the switching rates (number of switching events per cycle) generally change throughout the simulation, since the *trans* and *cis*-azo fractions do not remain constant. For $p_{tc}\gg p_{ct}$, the switching rates decrease severely over time since the fraction of *trans*-azo groups eventually becomes strongly diminished. By contrast, the switching rates remain high for $p_{tc}=p_{ct}$ because in the PSS *trans* and *cis* azobenzene are present in large numbers with $x_{t}\approx x_{c}\approx 0.5$.

### 3.3. Intermolecular Energy of TrisAzo Stacks before and during Light Irradiation

The structural changes of the columnar *TrisAzo* stacks are connected with energetic changes. Here, we examine how the intermolecular interactions between the stacked monomers change over time to learn about the causes of the structural degradation.

#### 3.3.1. Intermolecular Energy of the Cluster in Equilibrium (Initial State)

Let us first consider the intermolecular energies within the stack in thermal equilibrium, which is the reference state before light irradiation. The different parts of the stacked molecules contribute in different proportions to the binding of the cluster, see Figure 10 and Ref. [41]. In particular, the intermolecular (or binding) energies of the clusters are dominated by the dispersion interactions (π-π stacking) between the azobenzene arms of directly neighboring molecules. Other substantial contributions to the binding energies stem from dispersion interactions between the opposing BTA cores and between the azo arms and BTA cores. Hydrogen bonds between these central groups play a crucial role in stabilizing the column-like shape of the stacks, even though their energetic contributions are not predominant.

Here, the intermolecular energy between directly adjacent molecules, $E_{\mathrm{neigh}}^{\mathrm{inter}}$, is considered in particular. This measure generally differs from the total intermolecular energy of the full cluster, $E_{\mathrm{tot}}^{\mathrm{inter}}$. The latter includes also energetic interactions between molecules that are not direct neighbors. While $E_{\mathrm{tot}}^{\mathrm{inter}}$ is still the most relevant energy describing the cohesion of the *TrisAzo* aggregate, the energy $E_{\mathrm{neigh}}^{\mathrm{inter}}$ provides information about changes that are specific to the stacked cluster arrangement. Interestingly, in the case of the almost perfectly ordered equilibrated *TrisAzo* clusters considered here, both measures of intermolecular energy are nearly identical (Figure 10). Even the particular energetic contributions are present in nearly the same proportions with only slightly different values.

For the following discussion, the intermolecular energies $E_{\mathrm{neigh}}^{\mathrm{inter}}$ of the neighboring pairs are not only decomposed into different contributions, as inspired by the approach of Ref. [85]—see Ref. [41] for details of the energy calculation and decomposition. On top of that, the directly adjacent pairs of *TrisAzo* within the stacks are divided into two categories: defects and non-defects. This classification is based on whether the defect detection algorithm (see Appendix A) determines a defect between a particular monomer pair. The defect detection is carried out during the last 50 cycles of the MD trajectory, i.e., on the final 10% of simulation time.

#### 3.3.2. Intermolecular Energy upon Application of the CPM-B Approach

Figure 11 shows how the intermolecular energies between directly adjacent monomer pairs evolve during the CPM-B simulation with $p_{ct}$ = 5.0%. In the case of neighboring pairs with defects, the two most attractive energies, i.e., the azo–azo and the BTA–BTA contributions, become significantly weakened over time (Figure 11a). Each contribution is almost halved at the end of the 500 cycles. On top of that, the azo–azo curve is affected either slightly before or simultaneously with the BTA–BTA curve. Hence, these energy terms reflect the observation that the initial stacking of the azo groups is in some cases disrupted due to photoisomerization. Moreover, this effect precedes or coincides with a separation of the stacked BTA cores. The change in these two energetic contributions is mirrored in the diminished van der Waals energies (Figure 11b). As the BTA cores separate or become misaligned, the H-bonding energies and the number of H-bonds (Appendix A) are reduced as well.

The other energetic contributions are only slightly affected or do not change at all, such as the Coulomb term. The BTA–DMA and DMA–DMA contributions remain close to zero. Furthermore, the azo–DMA contributions are only weakened towards the end of the simulation run—possibly, when the misalignment of the monomers becomes very strong. The azo–BTA contributions fluctuate around a constant value and become the largest energy term at the end of the run. A possible reason is that upon the formation of defects, the loss of interactions between a BTA core with all three neighboring azo groups is compensated by increased interactions with only one or two of these azo arms.

When considering the neighboring monomer pairs without defects (non-defects) in the CPM-B, all energy terms stay nearly constant over time, see Figure 11c,d. This corresponds to the observation that the sections between the defects remain in a well-aligned stacked order. The azo–azo and BTA–BTA contributions become slightly weaker only near the end of the simulation, similar to the van der Waals and H-bonding energies. Possibly, this is when some of the monomer pairs are starting to develop towards (future) defects.

#### 3.3.3. Intermolecular Energy upon Application of the CPM-E Approach

For the CPM-E simulation with $p_{tc}$ = 1.88% and $p_{ct}$ = 0.12%, the time evolution of the intermolecular energies is shown in Figure 12. First, the neighboring pairs with defects are considered (Figure 12a,b). The weakening of the azo–azo and BTA–BTA contributions—and hence, of the van der Waals and H-bonding contributions—now proceeds much more strongly during the early stage of the simulation. This is also reflected in the strong reduction of the number of H-bonds (Appendix A). Here, the azo–azo and BTA–BTA energies are reduced even further than in the previously discussed CPM-B simulation, now to about a third of their initial value. Like in the CPM-B case, the change in both energies is temporally correlated. These energetic changes are a consequence of the accumulation of *cis*-azo groups in the cluster. Since the *cis* isomers persist for a longer time in the CPM-E simulations, the stacking of the azo arms is broken more efficiently. Other developments in the energetic contributions are similar to the CPM-B simulation. For instance, the BTA–DMA and DMA–DMA contributions stay close to zero and the Coulomb interactions overall remain unaffected. The azo–BTA and azo–DMA contributions strongly fluctuate but they generally remain near their initial values.

Second, the neighboring pairs without defects are considered. The CPM-E simulation shows a strong reduction in the azo–azo contribution to about half of its initial value (Figure 12c). This verifies that the longer persistence of the *cis*-azo isomers strongly weakens the azo–azo term since this effect was not observed in the CPM-B simulations (Figure 11c). Consequently, the van der Waals term is weakened in the CPM-E simulation as well (Figure 12d). As in the CPM-B case, the other energy contributions remain about constant or are only slightly affected—in particular, the BTA–BTA and the H-bonding term. The BTA–BTA and azo–BTA contributions are sufficiently attractive to keep these monomer pairs in stable and well-aligned stacked conformations. Hence, no defects are developed in these parts of the stack.

Appendix A show the energy decompositions for a large selection of CPM-B and CPM-E simulations at the end of each simulation. These results demonstrate that the same light-induced energetic changes in the stacks can be found for a wide range of switching probabilities (for both models, for monomer pairs with and without defects). Furthermore, Appendix A shows $E_{\mathrm{tot}}^{\mathrm{inter}}$ for the same simulations as discussed above. The total intermolecular energies of the entire cluster (not only between adjacent molecules) become slightly stronger during the CPM-B and CPM-E simulations. The now weaker interactions between previously neighboring monomers are overcompensated by stronger attractive interactions with other molecules. In particular, the azo–BTA and azo–DMA contributions are amplified and the azo–azo and BTA–BTA contributions are not as strongly reduced when considering $E_{\mathrm{tot}}^{\mathrm{inter}}$.

### 3.4. Mechanism of Defect Formation

After considering the structural and energetic changes of the *TrisAzo* stacks, we propose a mechanism by which the light-induced degradation of the cluster structure proceeds. Moreover, we point out which factors promote the observed defect formation.

Recall that the decomposition of the intermolecular energies has revealed the strongest attractive intermolecular energy contributions between the stacked monomer pairs in equilibrium (Figure 10). These are the azo–azo, BTA–BTA, and azo–BTA interaction (in this order). The results of the CPM simulations make it plausible, that the light-induced generation of defects in the *TrisAzo* stack follows a similar hierarchy. In other words, to separate the π-π-stacked, and hydrogen-bonded BTA groups at the center of the column, the binding between the azobenzene arms at the periphery has to be weakened first—especially because the azo groups are the most strongly bound. This is exactly what happens in the simulations upon photoisomerization. The stacking of two opposing *trans*-azo groups breaks when at least one of them switches to *cis*. If the *cis* groups persist for a longer time, the destabilization effect is retained as well.

Further perturbations due to ongoing switching events can then break up the binding between the other parts, particularly the BTAs. In some cases, the BTAs remain stacked and well-aligned, whereas in other cases, defects begin to develop between them. The defect formation is accompanied by the breaking of the hydrogen bonds. It is difficult to confirm the exact temporal progression of these effects with statistical data aside from Figure 11a and Figure 12a. However, it is plausible that the rupture of the H-bonds follows after the azo–azo interactions are sufficiently weakened.

What remains unclear and may be a topic for further research, is why some neighboring pairs in the stack develop into defects and others do not. Possibly, their conformations are offering specific weak points throughout the simulations. Despite the detailed simulation data available, it was not yet possible to extract a concrete criterion for the generation of a defect. Open questions remain, for instance, whether a certain number of azo groups must have switched to *cis* or if the intermolecular energies must cross a critical value. At the start of the simulations, no obvious energetic differences are found between pairs that end up with defects (Figure 11a and Figure 12a) or without them (Figure 11c and Figure 12c).

Generally, it would be desirable to perform multi-trajectory MD simulations to substantiate the findings. However, the results of our single-trajectory MD simulations are already expected to be reproducible. We find for all considered parameter values (except the two smallest switching probabilities), that new defects are generated in a certain fraction of neighboring molecule pairs, while the other fraction of pairs remains well-aligned. Moreover, as mentioned before, the stacking alignment decreases and the number of defects increases roughly proportional to the applied light intensity, see Figure 7 and Figure 8. This reveals a systematic trend in the change of the cluster structure, which corroborates the reproducibility of the single MD runs. For the smallest switching probabilities ($p_{tc}=p_{ct}$ = 0.1% and $p_{tc}$ = 0.188%, $p_{ct}$ = 0.012%), the detected defects have already been present after equilibration and, thus, before applying the CPM. The lack of newly generated defects can be clearly attributed to the small fraction of *cis*-azo groups together with low switching rates. Repeating the simulations is, hence, expected to yield the same results or at maximum very few newly generated defects.

#### The Impact of *Cis*-Azo Accumulation vs. Energy Dissipation via Photoswitching

Using both variants of the CPM allows us to compare the outcomes of both approaches and to determine reasons for the observed differences. As mentioned in Section 2.4.1, the CPM-B approach does not involve a lasting accumulation of *cis*-azo groups. Thus, the effect of the photoswitching nearly only consists of generating motion and dissipating energy. By contrast, in the CPM-E simulations, a larger fraction of *cis* azobenzene can accumulate. Energy dissipation and motion are present as well. Comparing the results of the two CPM variants may therefore reveal how these different aspects of the cyclic photoisomerization influence the cluster structure.

Let us consider the total number of the switching events, i.e., the sum of all forward and backward photoisomerization events during a simulation (Figure 9). This number is proportional to the total amount of dissipated energy in the system because in the utilized model each forward or backward isomerization event dissipates the same fixed amount of energy (see Section 2.3). Figure 9 demonstrates that most of the CPM-B simulations involve a much larger number of switching events and, thus, higher average switching rates and a higher average energy dissipation per cycle than the CPM-E simulations. Nonetheless, the light-induced structural changes are not stronger in the CPM-B simulations. This implies that in the CPM-E case, fewer or less frequent switching events are needed than in the CPM-B case to evoke a comparable degradation of the stacks. For instance, CPM-E simulations with $p_{tc}=p_{ct}$ = 1.00% or $p_{tc}$ = 1.88%, $p_{ct}$ = 0.12% result in similar changes of Δr(t) and Ψ(t) as the CPM-B run with $p_{tc}$ = 5.0%, $p_{ct}$ = 100.0% (Figure 8). In the latter case, more than 5000 switching events occur throughout the simulated period, while in the two CPM-E examples, this number is below 500 (Figure 9). The amount of dissipated energy in these simulations differs proportionally.

Furthermore, in the CPM-E simulations, a lower rate of photoisomerization is sufficient to induce a first noticeable change in the cluster structure. Note also that CPM-E simulations with intermediate switching probabilities ($p_{tc}<$ 5.0%) yield very similar structural changes (Figure 8b,d). Here, the higher average switching rates for systems with $p_{tc}=p_{ct}$ have no apparent impact on the structural changes. By contrast, for the largest considered switching probabilities, $p_{tc}=p_{ct}$ = 5.0% and $p_{tc}$ = 9.4%, $p_{tc}$ = 0.6%, the former simulation yields a much stronger degradation, while also having much a higher average switching rate (Figure 9). Hence, when the binding of the stack is already weakened by a persistent fraction of *cis*-azo groups, a significant increase in the switching rate (and thus, in the energy dissipation) amplifies the generation of defects even further.

These differences between the CPM approaches suggest that the photoswitching process itself (the dissipation of energy and the associated motion) generates defects a lot less efficiently if it occurs without the simultaneous accumulation of *cis*-azo groups. In other words, the prolonged existence of *cis* azobenzene weakens the binding of the columnar cluster over a long time, while the individual switching events rather play the role of short-time perturbations. The combination of repeated switching and the presence of a large fraction of *cis*-azo groups yields the strongest degradation of the cluster structure. Both phenomena are expected to be present in a realistic scenario.

Recall that the CPM-E simulations reproduce the photoisomerization kinetics most realistically among the considered approaches. The CPM-E results, therefore, demonstrate a cluster degradation that could be comparable on small length and time scales to the processes involved in the experiments of Lee et al. [24,25]. A more realistic simulation approach will have to take into account large bundles of *TrisAzo-H* stacks, which are laterally attached, to approach the experimental system more closely. If the photoisomerization of azobenzene has a dependence on the available free volume, a localized accumulation of *cis* isomers could play a role in *TrisAzo* clusters as well. It is not excluded that this effect could lead to larger, locally amplified defects or even breaking of the stacks, as observed in Ref. [39].

## 4. Conclusions

In this work, we investigate the effects of UV–Vis light on column-shaped stacks of three-arm azobenzene stars (*TrisAzo* molecules) in water. This system is modeled in fully atomistic MD simulations to determine which structural changes occur on the molecular scale and what forces are driving these changes. UV–Vis light causes repeated and reversible photoisomerization of the azo groups in the *TrisAzo* molecules. Thus, we have implemented and extended a detailed stochastic model to simulate the cyclic photoisomerization of azobenzene, including the explicit simulation of the *trans*–*cis* isomerization. The cyclic photoisomerization model reproduces the photokinetics of this system in good qualitative and quantitative agreement with theoretical and previous experimental results, although with accelerated isomerization rates. In its current form, the algorithm is applicable to various irradiation scenarios (light intensities, wavelengths) and may be easily extended further. The cyclic photoisomerization model may therefore be applied in future atomistic simulation studies on azobenzene-containing systems, e.g., on the light-induced deformation of azo-polymers [21,22,23].

The simulated effects of UV–Vis light yield a degradation of the previous well-ordered columnar arrangement of the *TrisAzo* stack. However, none of the simulations have resulted in breaking or disassembly of the stack into separate fragments. Instead, over time the clusters lose their columnar shape in an order–disorder transition. In particular, in a fraction of the adjacent *TrisAzo* pairs, the molecules become shifted and misaligned with respect to each other. In between such defects remain sections of stacked monomers with a more regular arrangement. If the switching rates in the CPM models are very low, the well-ordered arrangement of the stacks stays unperturbed.

A new interpretation of the experimentally observed morphological changes is devised. The simulations indicate that the photoisomerization of azobenzene has a twofold effect on the supramolecular stack. On the one hand, photoisomerization leads to an accumulation of *cis*-azo groups in the system. They strongly weaken the attraction between neighboring *TrisAzo* molecules because the *cis* groups cannot remain in a stacked arrangement. On the other hand, the repeated photoisomerization events act on the stack like ongoing perturbations since they generate motion and dissipate energy. With the diminished binding strength at the periphery (weakened azo–azo interactions), the ongoing isomerization occasionally separates the monomers at their centers, leading to the defects.

The results of the present simulation study suggest that the light-induced melting of the needle-like structures of Refs. [24,25] proceeds without the detachment of single molecules or cluster fragments on the here considered length and time scales. Since the needle-like structures in the experiments are much larger than the here considered stacks (micrometer scale vs. nanometer scale), they are likely composed of several laterally attached or bundled stacks. Thus, their light-induced morphological transition may involve the detachment of cluster fragments from such bundles in a different mechanism. This scenario may be investigated in coarse-grained simulation studies. In this manner, the reversibility of the structural changes may be addressed as well, as this was not investigated in the current work. A thorough understanding of these structures may facilitate the design of photoresponsive gels and supramolecular polymers, possibly with self-healing properties.

## Figures and Tables

**Figure 1 molecules-26-07674-f001:**
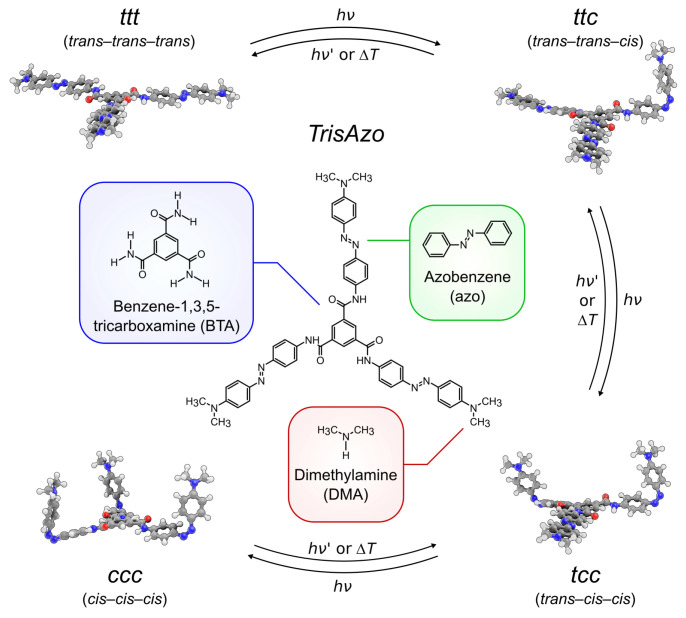
Chemical structure of *TrisAzo* (center) with its constituent groups highlighted in different colors. Around it, the four isomers of *TrisAzo* are shown as renderings from 3D models. The conformers of *tcc* and *ccc*, which have *cis*-azo arms pointing in different directions, are omitted. The molecular structures are visualized using a combination of the 3D graphics software Blender [49], the plugin BlendMol [50], and the MD visualization tool VMD [51].

**Figure 2 molecules-26-07674-f002:**
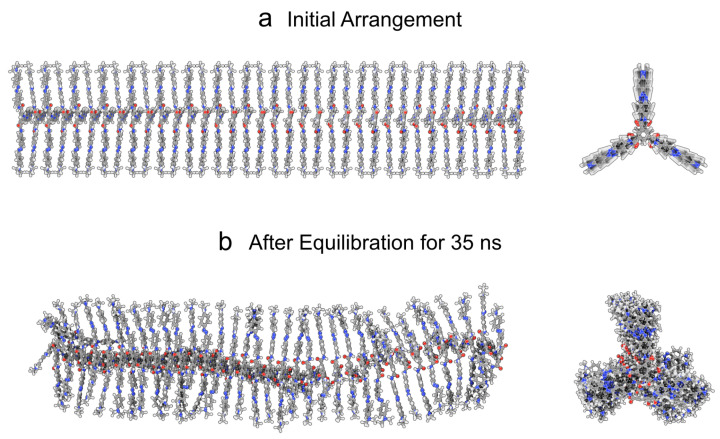
Columnar stack containing N=36
*TrisAzo* molecules shown from the side and top. (**a**) Initial arrangement of the stack before starting the simulations, and (**b**) after 35 ns of equilibration time under normal conditions. The visualizations are generated using a combination of the 3D graphics software Blender [49], the plugin BlendMol [50], and the MD visualization tool VMD [51].

**Figure 3 molecules-26-07674-f003:**
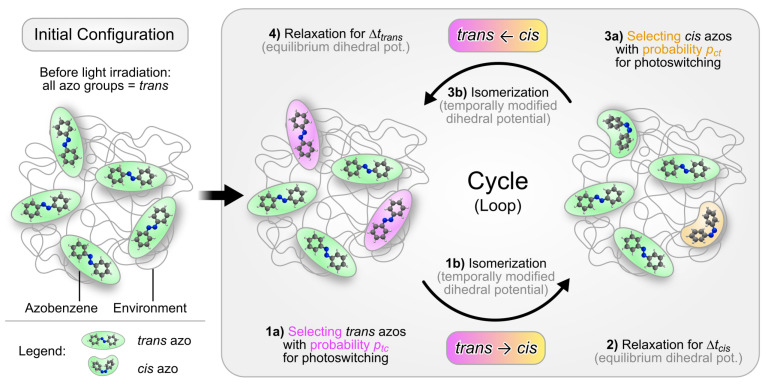
Illustration of the general sequence of events in the cyclic photoisomerization model. The system is composed of the light-reactive azobenzene groups and their environment (all photoinert parts of the system). The right-hand side depicts the simulation protocol for the photo-induced effects. Each photoisomerization cycle proceeds along steps 1–4 in a looped manner, repeated *N* times.

**Figure 4 molecules-26-07674-f004:**
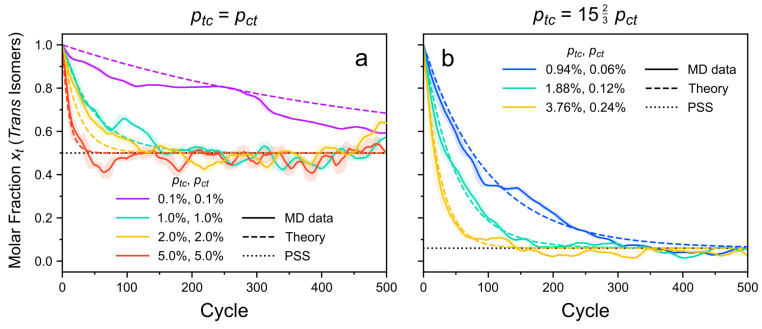
Molar fraction $x_{t}\left(t\right)$ of the *trans* isomers with respect to all azo groups as a function of time. The *cis* curves are omitted because $x_{c}\left(t\right)=1-x_{t}\left(t\right)$. (**a**) $p_{tc}=p_{ct}$. (**b**) $p_{tc}=15\frac{2}{3}p_{ct}$. This highly asymmetrical ratio of switching probabilities is inferred from previous experiments of Kind et al. [83]. Colors indicate different values of $p_{tc}$ and $p_{ct}$. MD data are represented by solid lines. Each MD curve is smoothed via a running average (window size of 20 cycles, corresponding to 0.96 ns). Shaded error bands indicate the standard deviation of the raw curve in each window. The calculated time evolution of the molar fractions is depicted as dashed lines and the respective steady-state molar fractions are shown as dotted lines.

**Figure 5 molecules-26-07674-f005:**
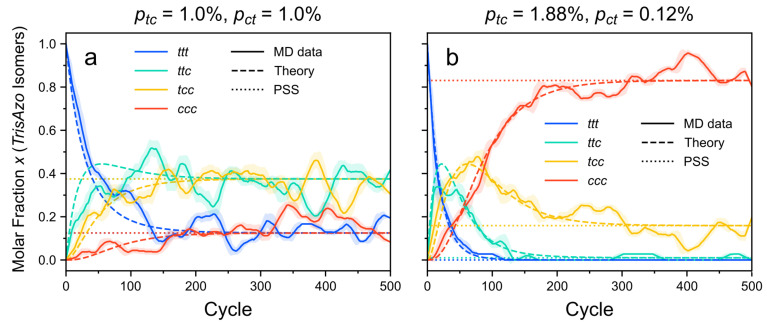
Molar fractions of the four different *TrisAzo* isomers with respect to all *TrisAzo* molecules as a function of time. (**a**) $p_{tc}=p_{ct}$. (**b**) $p_{tc}=15\frac{2}{3}p_{ct}$, like in the experiments of Kind et al. [83]. MD data are represented by solid lines. The meaning of the line styles and the settings of the running average are the same as in Figure 4.

**Figure 6 molecules-26-07674-f006:**
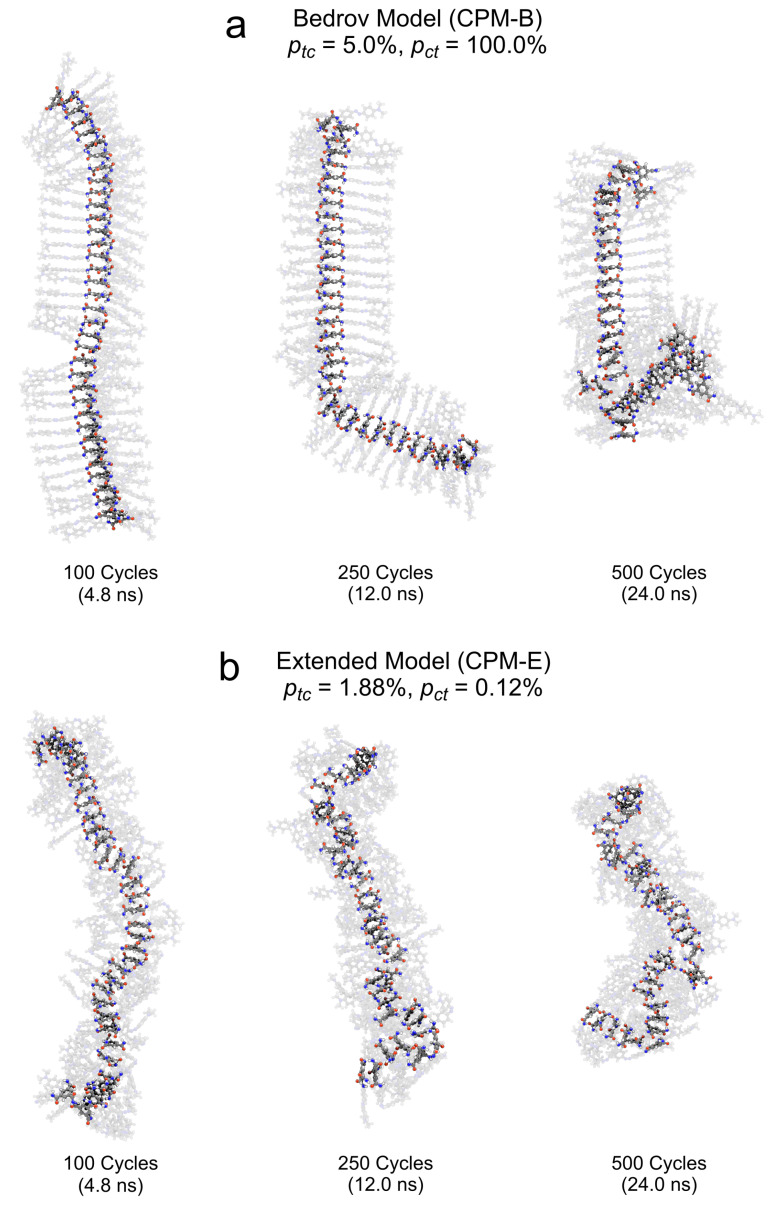
Simulation snapshots of the *TrisAzo* stack throughout the application of the cyclic photoisomerization model. The images illustrate the time evolution of the induced structural changes in the *TrisAzo* stacks in the (**a**) CPM-B and (**b**) CPM-E approach. The clusters are shown at an early stage (100 cycles), at half time (250 cycles), and at the end of the simulation runs (500 cycles). The BTA groups are highlighted for better visibility. The visualizations are created using the 3D graphics software Blender [49], the plugin BlendMol [50], and the MD visualization tool VMD [51].

**Figure 7 molecules-26-07674-f007:**
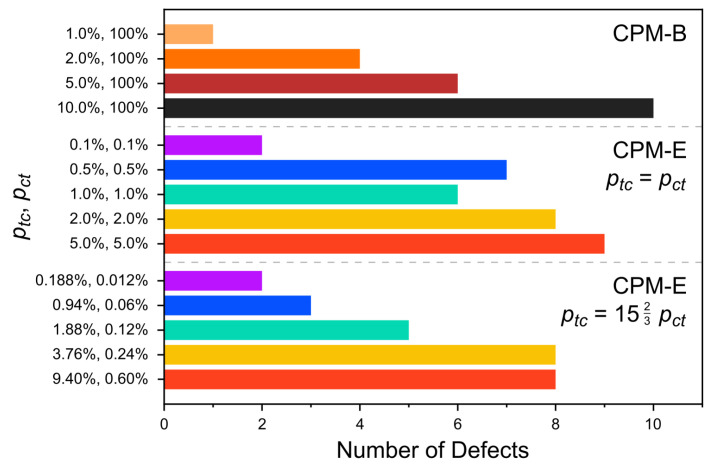
Number of defects in the cluster during the final 50 of 500 cycles (last 2.4 ns of 24 ns in total) of each simulation run. Results for different switching probabilities both in the CPM-B and CPM-E approach are shown. The defect detection algorithm is described in the Appendix A.

**Figure 8 molecules-26-07674-f008:**
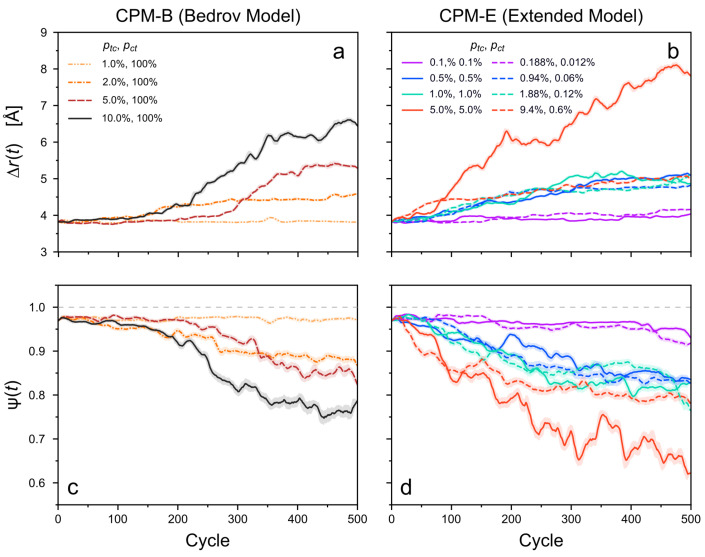
(**a**,**b**) Average pairwise COM distance between adjacent monomers, Δr(t), and (**c**,**d**) average global columnar orientation order parameter of the *TrisAzo* cluster, Ψ(t), each as a function of time. Results for the CPM-B (left-hand side) and the CPM-E (right-hand side) are shown for different switching probabilities per cycle. Each curve is smoothed via a running average (window size of 20 cycles, corresponding to 0.96 ns). Shaded error bands indicate the standard deviation of the raw curve in each window. The gray dashed line indicates the maximum possible value of Ψ(t).

**Figure 9 molecules-26-07674-f009:**
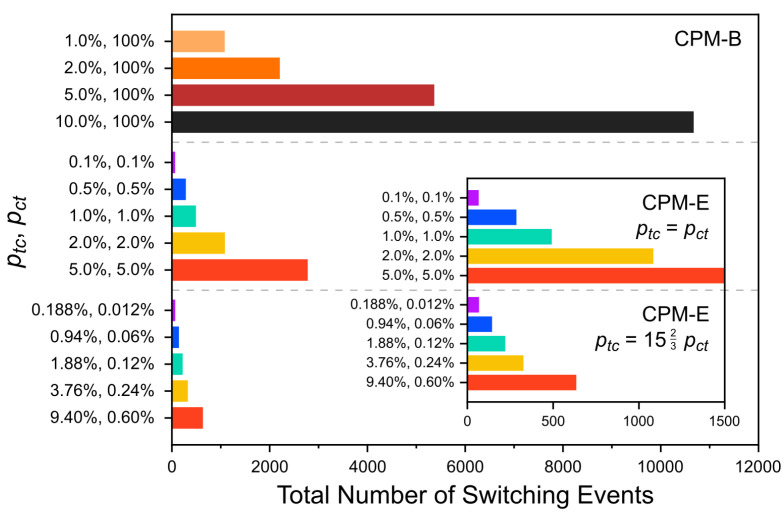
Total number of photoisomerization events (forward and backward) throughout each simulation (over 500 cycles) for different switching probabilities in the CPM-B and CPM-E approach. The inset zooms in on the number of switching events in the CPM-E simulations.

**Figure 10 molecules-26-07674-f010:**
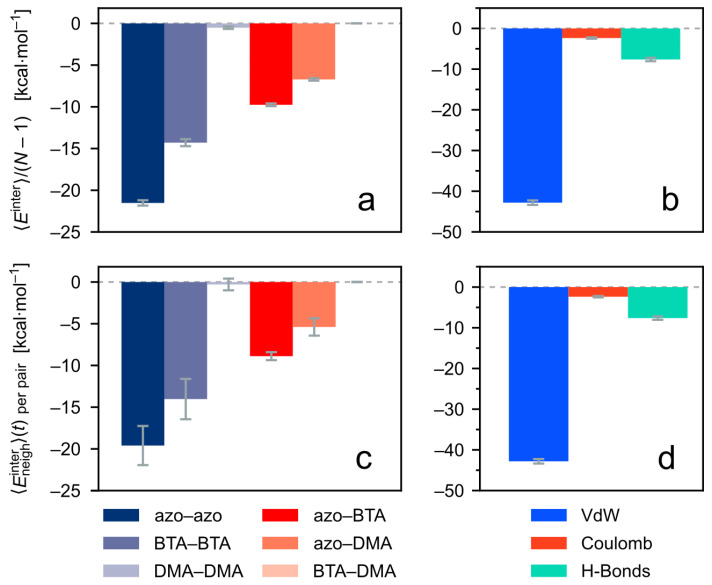
Decomposition of the intermolecular energies of the *TrisAzo* cluster with N=36 in the equilibrium state, i.e., in the initial state before starting the CPM simulations. (**a**,**b**) Total intermolecular energies of the cluster divided by the number of neighboring monomer pairs. (**c**,**d**) Average intermolecular energies between directly adjacent monomer pairs. Left-hand side: energy decomposition based on the interactions between the different physical parts of *TrisAzo*. Right-hand side: energy decomposition based on the relevant non-covalent interactions between the molecules.

**Figure 11 molecules-26-07674-f011:**
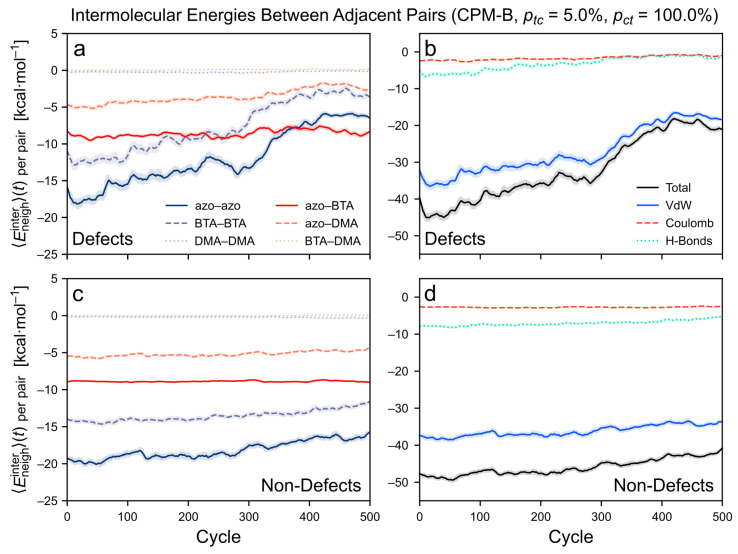
Intermolecular energies between adjacent monomer pairs in the *TrisAzo* stack as a function of time. The CPM-B simulation with $p_{tc}$ = 5.0% is shown. The energy contributions are averaged for (**a**,**b**) all neighboring monomer pairs with defects, and (**c**,**d**) without defects. Left-hand side: energy decomposition by interactions between the different physical parts of the *TrisAzo* molecules. Right-hand side: energy decomposition by types of non-covalent interactions. Each curve is smoothed via a running average (window size of 20 cycles, corresponding to 0.96 ns). Shaded error bands indicate the standard deviation of the raw curve in each window.

**Figure 12 molecules-26-07674-f012:**
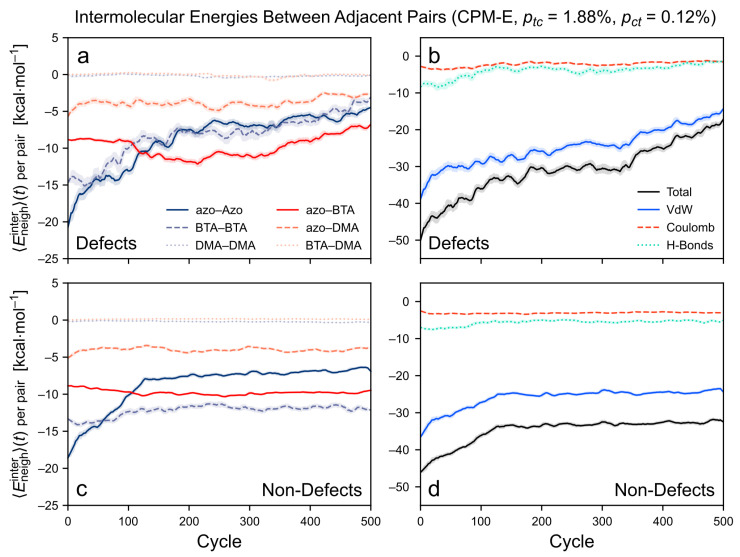
Time evolution of the intermolecular energies between the adjacent monomer pairs in the *TrisAzo* stack for the CPM-E simulation with $p_{tc}$ = 1.88%, $p_{ct}$ = 0.12%. The energy contributions are averaged for (**a**,**b**) all neighboring monomer pairs with defects, and (**c**,**d**) without defects. Left-hand side: energy decomposition by interactions between the different physical parts of the *TrisAzo* molecules. Right-hand side: energy decomposition by types of non-covalent interactions. Each curve is smoothed via a running average (window size of 20 cycles, corresponding to 0.96 ns). Shaded error bands indicate the standard deviation of the raw curve in each window.

## Data Availability

The code and tools for the cyclic photoisomerization model are available online at https://github.com/Markus91Koch/CyclicPhotoisomerizationModel accessed on 17 December 2021.

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
