# Peer review of "Cyclic Photoisomerization of Azobenzene in Atomistic Simulations: Modeling the Effect of Light on Columnar Aggregates of Azo Stars"

_molecules, 2021, doi:10.3390/molecules26247674_

Round 1
Reviewer 1 Report
This manuscript discussed the driving force of the needle-like structures “melting under light” phenomena by using force field level atomistic simulation. The phenomena are interesting and still under debate. The most difficult part is to incorporate photoisomerization during the simulation. Although the photoisomerization of azobenzene is a quantum chemistry topic, the author used their model which is modified from a stochastic model, CPM (Cyclic Photoisomerization Model), to simulate the effect of light. This is a part of authors’ series works. I noticed that authors have already researched TrisAzo’s properties by using DFT calculations and the self-assemble behavior of TrisAzo molecules in water (Molecules 2019, 24(23), 4387). This paper focuses on the light-induced loses of columnar shape. And it based on Markus Koch’s dissertation.
This is a high-quality manuscript. The figures are clear and pretty. And I have a smooth reading experience during the reviewing. Meanwhile, the CPM code is also available on GitHub. The authors do provide theoretical insights into the microscopic order-disorder mechanisms of azo stars Columnar stacks. The computational results can support their major conclusions. It’s my great pleasure to recommend this work for publication after tiny reversions.
In addition, I have several comments and suggestions about this work. Firstly, it would be better if you set up a larger system. For example, a system with thousands of azobenzene star molecules. Or you can build a system with 7-20 bundled of TrisAzo columnar stacks. Larger system simulation may provide new insights. LAMMPS can achieve that. Secondly, the connection between macroscopic phenomenon, “melting under light” and microscopic simulation results is not very strong. The mesoscopic discussions are more convincing. As you said in last paragraph, coarse-grained simulation is needed.
Suggestion 1:
Figure 1,2 and 6:
The molecular structures are plotted with very high quality. It seems like that you are using ChimeraX or PyMOL. Most molecular visualization software has citation requisition, although it’s not mandatory. Could you please cite their software for demonstrating the value of our work?
Suggestion 2:
Please add a discussion about the reproducibility of your MD and CPM-E simulation. Have you used multi-trajectory simulation? Multi-trajectory MD is not needed if you can confirm that you have sufficient sampling.
Is the result “some neighboring pairs in the stack develop into defects and others do not” reproducible? (page 19, line 632)
Reviewer 2 Report
The present paper shows a computational study of the influence of light on the structure and stability of the columnar stacks of three-arm azobenzene stars.
The analysis is based on the fully atomistic molecular dynamics. This is an interesting work.
I believe that this paper will initiate more similar studies.
The authors cite on several places Ref. 41, which seems to be unpublished work also submitted to Molecules. It would have been much more convenient if the referee could have access to this paper.
Anyway, this is a high-quality work, and I would recommend it for publication.
